# Response of Anaerobic Granular Sludge Reactor to Plant Polyphenol Stress: Floc Disintegration and Microbial Inhibition

Shilin Bi [1,†], Hua Lian [1,†], Huiya Zhang [1], Zexiang Liu [1], Yong Chen [1] and Jian Zhang [1,2,*]

[1] Guangxi Key Laboratory of Clean Pulp & Papermaking and Pollution Control, College of Light Industry and Food Engineering, Guangxi University, Nanning 530004, China; 15650079435@163.com (S.B.); lihua321_wen@126.com (H.L.); zhang_huiya@163.com (H.Z.); liuzexiang314@163.com (Z.L.); 2216391003@st.gxu.edu.cn (Y.C.)

[2] Anhui Bossco Environmental Protection Technology Co., Ltd., Nanning 530007, China

[*] Correspondence: zhangjian_gx@gxu.edu.cn

[†] These authors contributed equally to this work.

**Abstract:** Plant polyphenols are potential inhibitors for the anaerobic treatment of wastewater from the wood processing, pharmaceutical, and leather industries. Tannic acid (TA) was selected as a model compound to assess the inhibitory effect of plant polyphenols in simulated wastewater in this study. The influences of TA on methanogenic activity, sludge morphology, and the microbial community were investigated under glucose and sodium acetate as carbon substrates, respectively. The results show that a threshold concentration of TA above 1500 mg·L$^{-1}$ that triggers significant methanogenesis depression and volatile fatty acids (VFAs) accumulation. In addition, granules might be weakened by TA addition, reflected in changes in extracellular polymeric substances (EPS) within the granules and an increase in floc in the effluent. The anaerobic granular sludge (AnGS) fed with sodium acetate was more sensitive than the presence of glucose as the substrate when facing the challenge of TA. The concentration of the *mcrA* gene in granular sludge decreased markedly in response to TA stress, providing direct evidence that a high concentration of TA caused the inhibition of specific gene expressions. This study provides details about the adverse impacts of TA stress on methane production, the microbial community, and granule integrity, deepening our understanding of the anaerobic treatment of plant polyphenols contained in wastewater.

**Keywords:** anaerobic digestion; inhibition; plant polyphenols; plant refining; wastewater





## 1. Introduction

Anaerobic technology is commonly employed for treating concentrated organic waste due to its effectiveness, affordability, and applicability. The leather and wood industries generate significant volumes of concentrated organic effluent containing plant polyphenols [1,2]. Hydrolyzed tannins (oligomeric tannins) and condensed tannins (high-molecular-weight tannins) are the main types of plant polyphenols [3]. The shared phenolic hydroxyl groups in plant polyphenols may disrupt microbial metabolism by interacting with cell proteins, polysaccharides, and metal ions [4,5]. Hence, plant polyphenols, also known as "tannin", can potentially affect the anaerobic digestion of tannin-containing wastewater [6].

Tannins exert negative impacts on enzyme activity by creating hydrogen bonds with the active areas of enzymes [7,8]. The methanogenic toxicity of TA was reported to depend on concentration, and its degradation would constantly cause metabolic depression until the TA was completely degraded [9]. But a high initial concentration of tannin will interfere with the microbial degradation of tannin itself [10]. The methanogenic inhibition caused by tannins is determined by their molecular weight or degree of polymerization. For example, oligomeric tannins exhibit greater methanogenic toxicity compared to tannin monomers and high-molecular-weight tannins. This is because tannin monomers have a lower binding affinity to enzymes than oligomeric tannins. And high-molecular-weight tannins typically

encapsulate their phenolic hydroxyl groups within their molecules, thereby reducing accessibility for enzymes [11]. Depression caused by tannins is not only confined to disrupting extracellular coenzyme activity during anaerobic digestion. The concentration of coenzyme F420 in anaerobic sludge declined as the concentration of TA increased [12]. Furthermore, methyl coenzyme-M reductase (MCR) is a key enzyme of methanogenesis [13]. The level of the *mcrA* gene, which encodes the MCR subunit, is positively correlated with methanogen quantification [14]. Analysis of the *mcrA* gene has been widely used in many anaerobic digestion studies to evaluate sludge methanogenic activity [15,16]. However, the *mcrA* gene has rarely been mentioned in studies investigating the methanogenic inhibition of tannins.

The impacts of tannin on methanogenic toxicity and biodegradability have been discussed the most. However, the impacts of TA-containing wastewater on the morphology and microbiological population of anaerobic granular sludge have rarely been explored. The excellent stability and reliability of anaerobic granular sludge-based reactors depend on the condensed granule, which grows as a spherical aggregate under hydraulic selection. Extracellular polymeric substances (EPS) secreted by cells are the key substances that guarantee density and stability by promoting the adhesion of microorganisms to form dense biofilms [17]. The hydrophobic interaction between tannin and EPS (e.g., proteins and polysaccharides) might disturb cell aggregation and granule formation [18]. Tannin can have intricate effects on anaerobic digestion, which includes sequential processes such as hydrolysis, acidification, acetogenesis, and methanogenesis. Plant polyphenols hamper methane synthesis, altering the evolution of microbial communities [19]. Microbial community domestication is guided by specific substrate provisions [20,21]. Hence, biodegradable tannin, which acts as a co-substrate in anaerobic digestion, can alter the microbial community [22].

Tannin poses a risk to anaerobic reactors treating wood and leather-processing wastewater. Specific attention is required for the deactivation of methanogenic organisms, the disintegration of granules, and changes in microbial composition in an anaerobic reactor under tannic stress. Therefore, the oligomeric tannin (tannic acid, TA) was selected to simulate the plant polyphenols. This study monitored the changes in methanogenic activity, sludge structure, and microbial composition of anaerobic granular sludge (AnGS) in laboratory reactors supplied with sodium acetate and glucose as substrates.

## 2. Materials and Methods

### 2.1. Inoculum and Substrates

The AnGS used in this study was obtained from the up-flow anaerobic sludge blanket (UASB) for wastewater treatment in Guangxi, China. In this experiment, a size range of 0.5 to 2.5 mm of granule sludge was selected. In addition, the ratio of volatile suspended solids (VSS) to total suspended solids (TSS) was 0.78.

In this experiment, artificial wastewater was used as the substrate. Glucose and sodium acetate were used as carbon sources in artificial wastewater, respectively. Additionally, $NH_4Cl$ and $KH_2PO_4$ were used as nitrogen sources and phosphorus sources, respectively. The ratio of the chemical oxygen demand (COD):N:P in the artificial wastewater was 500:5:1. The sludge microorganisms were provided with mineral elements and trace elements to facilitate their optimal growth. The trace metal solution ($mg \cdot L^{-1}$) was as follows: $FeSO_4 \cdot 7H_2O$, 3.00; $CuCl_2 \cdot 2H_2O$, 0.50; $NiCl_2 \cdot 6H_2O$, 0.25; $CoCl_2 \cdot 6H_2O$, 1.34; $MnCl_2 \cdot 2H_2O$, 0.54; $AlCl_3$, 0.05; $H_3BO_3$, 0.10; $ZnCl_2$, 0.05; $(NH_4)_6Mo_7O_{24} \cdot 4H_2O$, 0.05; $Na_2WO_4 \cdot 2H_2O$, 0.05. The mineral elements ($mg \cdot L^{-1}$) were as follows: $CaCl_2$, 22.47; $MgCl_2 \cdot 6H_2O$, 21.86.

Tannic acid (TA, molecular formula: $C_{76}H_{52}O_{46}$, AR), bovine serum albumin (BAS, standard), and glucose (standard) were purchased from Aladdin Chemistry Reagent Co., Ltd. (Shanghai, China). The Coomassie Brilliant Blue dye was obtained from Merck Drugs and Biotechnology, Darmstadt, Germany. In addition, all other reagents (AR) were sourced from Macklin Chemicals Co., Ltd. (Shanghai, China). AR (Analytical Reagent) is the purity specification of chemical reagents.

## 2.2. Experimental Design and Operation

Before the experiment, the inoculum was activated using specific wastewater. The activation process is detailed in Text S1. The experiments were conducted in 8 identical serum bottles with a total working volume of 250 mL, comprising 160 mL of artificial wastewater and 40 mL of inoculum. In the sodium acetate group, sodium acetate served as the sole carbon source substrate. The control group had no TA addition, while STA-I/II/III/IV contained TA at concentrations of 500, 1000, 1500, and 2000 mg·L$^{-1}$, respectively. Conversely, glucose was the sole carbon source substrate in the glucose group. Similar to the sodium acetate group, the control had no TA addition, while GTA-II/IV contained TA at concentrations of 1000 and 2000 mg·L$^{-1}$, respectively. The equivalent COD of glucose and sodium acetate was 8000 mg·L$^{-1}$ in their respective groups. Moreover, the specific COD of 1 g of TA was measured to be 1.024 g. To mitigate rapid acidification, NaHCO$_3$ was added to the influent in the glucose group, maintaining a NaHCO$_3$:COD mass ratio of 0.8:1. Detailed operational and influent parameters for each reactor are presented in Table 1. After purging with pure N$_2$ for 5 min to maintain an anaerobic environment, all reactors were placed in a water-bath shaker set at 135 rpm and 36.5 ± 0.5 °C for anaerobic digestion (AD) experiments. Daily methane production was measured using an automated methanogenic potential testing system (AMPTA3, Nova Skantek, Beijing, China). Additionally, a 3% NaOH solution was connected at the exit to neutralize other acidic gases. To ensure steady-state operation under identical conditions, TA was not added during the initial three days. On the 4th day, TA was introduced to the corresponding reactors.

**Table 1.** Operational and influent parameters in the reactor for 54 days.

| Index | Unit | Sodium Acetate Group | | | | | Glucose Group | | |
|---|---|---|---|---|---|---|---|---|---|
| | | S-CON | STA-I | STA-II | STA-III | STA-IV | G-CON | GTA-II | GTA-IV |
| HRT [a] | h | 24 | 24 | 24 | 24 | 24 | 24 | 24 | 24 |
| TA [b] | mg·L$^{-1}$ | 0 | 500 | 1000 | 1500 | 2000 | 0 | 1000 | 2000 |
| Substrate COD [c] | mgCOD·L$^{-1}$ | 8000 | 8000 | 8000 | 8000 | 8000 | 8000 | 8000 | 8000 |
| NaHCO$_3$ | mg·L$^{-1}$ | ND | ND | ND | ND | ND | 6400 | 6400 | 6400 |

[a], hydraulic residence time; [b], tannic acid; [c], the equivalent chemical oxygen demand of carbon source substrate.

To ensure data stability and reliability, all experiments in this study were conducted in triplicate, with results reported as mean ± standard deviation. Throughout the research process, measurements of the soluble chemical oxygen demand (SCOD), volatile fatty acids (VFAs), TA concentration in the reactor effluent, and methane yield were conducted every two days. The extraction and determination of extracellular polymeric substances (EPS) from anaerobic granular sludge (AnGS) were performed at the beginning and end of the experiment for each anaerobic reactor. Additionally, microbial composition analysis was conducted on the sludge obtained from each anaerobic reactor at the end of the experiment, along with the inoculum.

## 2.3. Conventional Analysis Methods

VSS and TSS were tested following standard methods [23]. pH was measured using a bench-top acidimeter [24]. SCOD was determined using a standard potassium dichromate assay [25]. VFAs were quantified by gas chromatography (GC-8890, Agilent, Santa Clara, CA, USA) with a flame ionization detector (HP-INNOWax column) [26]. The TA content was evaluated through UV absorbance [27]. The methane yield was expressed as NmLCH$_4$·g COD$^{-1}$ [28]. The value of Specific Methane Activity (SMA) was obtained by fitting a one-dimensional linear model [29]. The particle size distribution of sludge was analyzed by a laser particle sizer (Mastersizer 2600, Malvern, Shanghai, China). The apparent morphology of AnGS was photographed through an industrial microscope (L200N, Nikon, Tokyo, Japan). The photos of the turbidity of the anaerobic effluent were taken with a portable camera (D-LUX 7, Leica, Shanghai, China) (see Text S2 for more details).

### 2.4. EPS Extraction and Analysis

A modified heat extraction method was used to obtain tightly bound EPS (TB-EPS) and loosely bound EPS (LB-EPS) from the inoculum and sludge samples [30]. In a 50 mL centrifuge tube, 4 g of granular sludge was thoroughly mixed with 30 mL of a 0.85% NaCl solution. After centrifugation at 8000 rpm for 20 min, the organic matter in the supernatant was collected as LB-EPS. The residual liquid was then discarded, and the sludge pellet left in the centrifuge tube was re-mixed with a 0.85% NaCl solution to restore its original volume of 30 mL. The mixture was heated in a water bath at 80 °C for 30 min, followed by centrifugation at 3200 rpm for 30 min. The organic matter in the supernatant was collected as TB-EPS.

According to the report, protein (PN) and polysaccharide (PS) were measured as the main examined components of EPS [31]. Therefore, the TB-EPS and LB-EPS fractions, filtered through a 0.45 μm acetate fiber filter, were immediately used to determine the concentration of PN and PS. PN was analyzed using the modified Bradford method, with bovine serum albumin (BSA) as the standard [32], while PS was analyzed using the anthrone–sulfuric acid method, with glucose as the standard [33].

### 2.5. Microbial Community and mcrA Gene Analysis

The 16S rRNA analysis was conducted via Illumina MiSeq sequencing by Majorbio Bio-pharm Technology Co., Ltd. (Shanghai, China). Before analysis, the inoculum and sludge samples collected in each reactor were stored at −80 °C. The sludge samples were obtained by mixing sludge from the parallel reactors. Specifically, 338F (5′-ACTCCTACGGGAGGCAGCA-3′) and 806R (5′-GGACTACHVGGGTWTCTAAT-3′) were applied as primers for archaea, and 524F10extF (TGYCAGCCGCCGCGGTAA) and Arch958RmodR (YCCGGCGTTGAVTCCAATT) were utilized as primers for archaea [34]. Primers for both bacteria and archaea were used for the amplification of the V3–V4 region of the 16S rRNA gene. Based on the similarity threshold used of 97%, sequences are clustered into bins called 'Operational Taxonomic Units' (OTUs) [35]. In this study, the analysis of the microbial community (bacteria and archaea) in sludge was based on OTUs. The mcrA gene content was quantified utilizing quantitative PCR (qPCR) technology, and mcrA-F (5′-GGT GGT GTM GGA TTC ACA CAR TAY GCWACA GC-3′) and mcrA-R (5′-TTC ATT GCRTAG TTW GGR TAG TT-3′) were used as primers [36]. The analysis of sequencing data was carried out on http://www.majorbio.com (accessed on 25 March 2024).

### 2.6. Statistical Analysis

Statistical analysis was performed using SPSS 26.0 software. Specific details are shown in Text S3.

## 3. Results and Discussion

### 3.1. Performance of AnGS under Different TA Concentrations

TA exhibits a more pronounced inhibitory effect on the organic removal capacity of AnGS when sodium acetate is used as a carbon source. As depicted in Figure 1, within the sodium acetate group, the conversion of sodium acetate to $CO_2$ and $CH_4$ takes place in the sludge according to reaction (3) (Table 2). This reaction simultaneously generates a significant quantity of inorganic carbon alkalinity in the aqueous system, thereby raising the pH within the reactor. In the sodium acetate group, $NaHCO_3$ was not added to maintain an alkaline environment in the reactor, following the methodology outlined by Dareioti et al. [37]. At TA concentrations below 1000 mg·L$^{-1}$, the SCOD removal and gas yield in STA-I and STA-II reactors were comparable to those in the control reactor (CON). However, a decrease in SCOD removal and a further reduction in bulk pH were observed in STA-I and STA-II after the 50th day, indicating a slight inhibitory effect of TA during the extended operation of the reactors. The bulk pH of both STA-I and STA-II was lower than that of the CON, attributed to the degradation of a portion of TA by hydrolytic acidifying

bacteria within the sludge coinciding with the occurrence of reaction (3) (Figure S1a). The resultant acidity from this process neutralized the alkalinity in the methanogenic process.

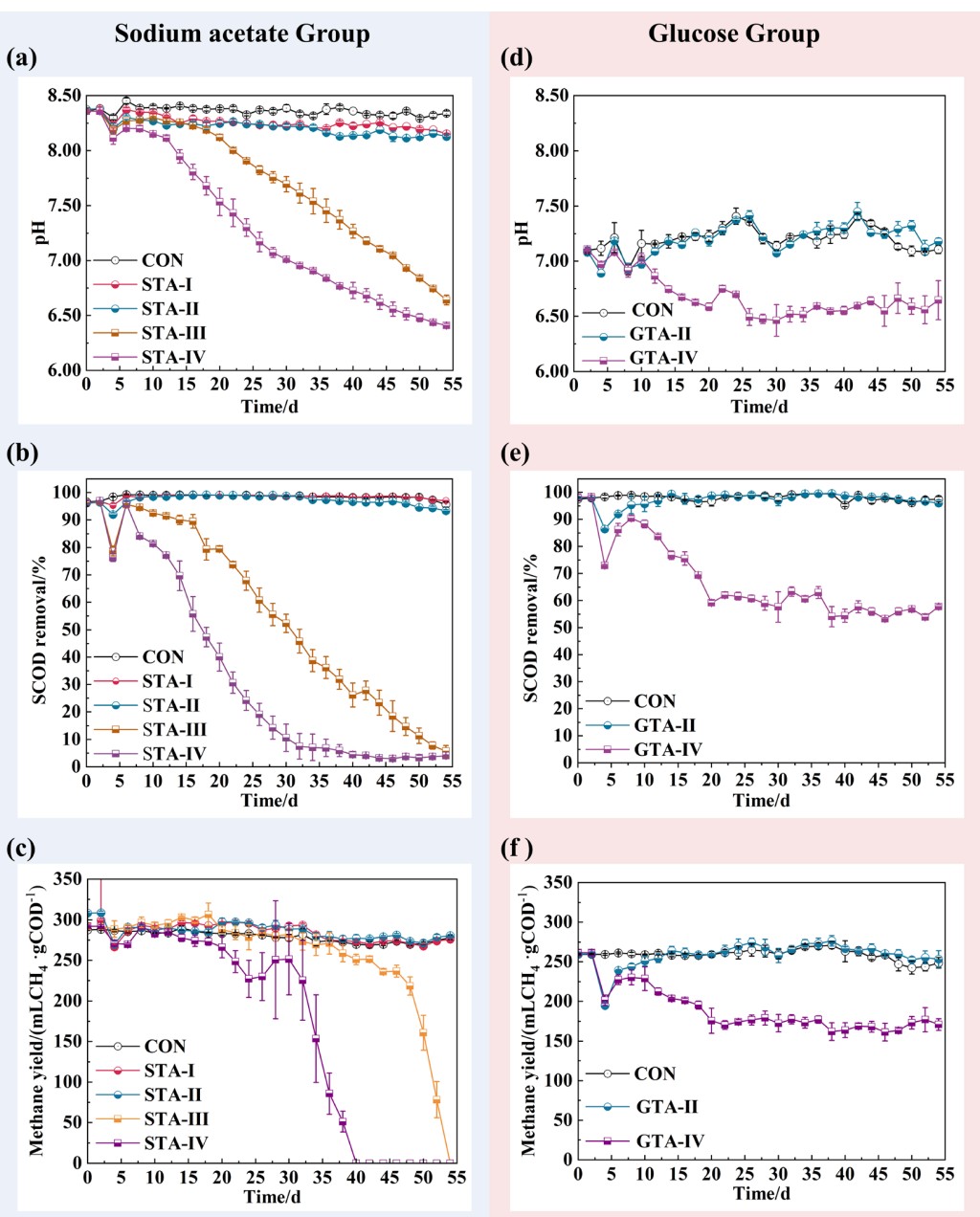

**Figure 1.** pH and SCOD removal in anaerobic effluent, and methane production in the reactor under TA stress. Sodium acetate group (**a**–**c**) on the left and glucose group (**d**–**f**) on the right.

**Table 2.** The main chemical and biological reactions example through glucose occurring in an anaerobic digestion system.

| Number | Reaction | Process |
|---|---|---|
| (1) | $x\mathrm{C_6H_{12}O_6} \rightarrow \mathrm{CH_3COO^-} + \mathrm{CO_2} + \mathrm{H^2} + \mathrm{H^+}$ | Acidification |
| (2) | $\mathrm{CO_2} + \mathrm{H_2O} \leftrightarrow \mathrm{H_2CO_3} \leftrightarrow \mathrm{HCO_3^-} + \mathrm{H^+}$ | |
| (3) | $\mathrm{CH_3COO^-} + \mathrm{H^+} \rightarrow \mathrm{CH_4} \uparrow + \mathrm{CO_2} \uparrow$ | Methanogenesis |
| (4) | $4\mathrm{H_2} + \mathrm{HCO_3^-} + \mathrm{H^+} \rightarrow \mathrm{CH_4} \uparrow + 3\mathrm{H_2O}$ | |

Meanwhile, the concentration of VFAs in the effluent ranged from 100 to 500 mg·L$^{-1}$ when the concentration of TA was below 1000 mg·L$^{-1}$ (Figure 2). The content of acetic acid in TA-I and TA-II reactors was consistent with that in the corresponding control reactor (CON). The inhibitory effect of TA and its breakdown products on the conversion of acetic acid to methane was inadequate. At TA concentrations of 1500 and 2000 mg·L$^{-1}$, STA-III and STA-IV reactors exhibited a gradual collapse trend after a brief steady state at the beginning of the experiment. According to the data presented in Figure S1, the removal of TA in STA-III and STA-IV remained consistently high, ranging from 75.12% to 85.52% over the entire experimental period. However, the removal of SCOD and the bulk pH significantly decreased, indicating a substantial hindrance to the sludge's capacity to degrade substances. By the 54th day, the sludge in STA-III had markedly diminished its capacity to degrade acetic acid due to the presence of TA, resulting in only 80 mg·L$^{-1}$ of acetic acid being converted. In STA-III, the pH of the system was below 7.00 as a result of the hydrolytic acidification of TA, leading to the collapse of acidification. The elimination of SCOD from STA-IV was less than 10.15% on the 30th day, and gas production ceased on the 40th day. These results indicate that TA and its degradation products seriously inhibit the methanogenesis of acetic acid conversion when the TA concentration exceeds 1500 mg·L$^{-1}$.

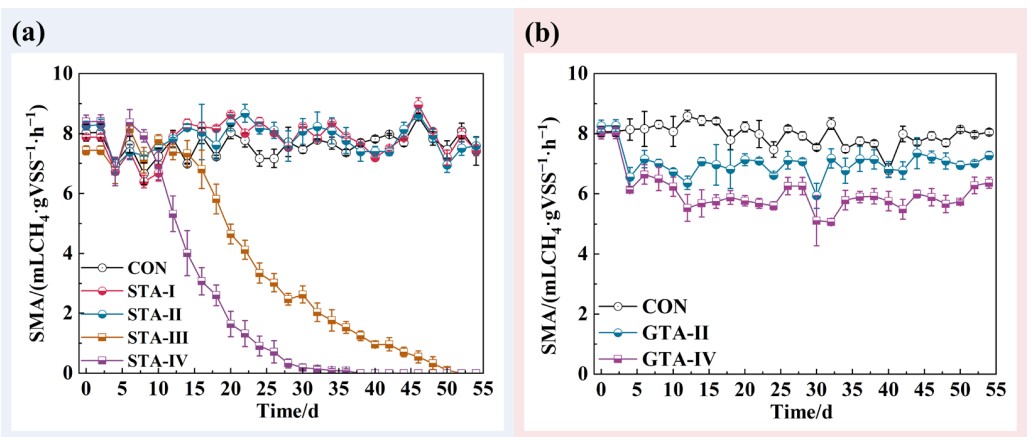

**Figure 2.** The SMA of AnGS under TA stress ((**a**) sodium acetate group, (**b**) glucose group).

To investigate the specific role of TA in the anaerobic degradation (AD) of AnGS, we examined the inhibitory effect of TA in the presence of glucose substrate at TA concentrations of 1000 mg·L$^{-1}$ and 2000 mg·L$^{-1}$. The pathway of glucose degradation by microorganisms in the anaerobic reactor is depicted in reactions (1) to (4) (Table 2). Since reaction (1) is the rate-limiting step, the significant proton production during glucose hydrolysis and fermentation caused the bulk pH of the glucose group to be close to neutral, markedly lower than that of the sodium acetate group, despite the addition of 6.4 g·L$^{-1}$ of NaHCO$_3$. The results showed that TA had no effect on the anaerobic digestion of glucose in the GTA-II reactor at a TA concentration of 1000 mg·L$^{-1}$. However, with an increase in TA concentration, a rapid decrease in bulk pH, SCOD removal, and gas production was observed in the GTA-IV reactor. Nevertheless, after one month, the microorganisms within the sludge gradually adapted to the toxicity of TA, maintaining SCOD removal at approximately 60.15%. The bulk pH in the GTA-IV reactor initially decreased and then gradually increased, with the reactor performance remaining stable until the end of the 54-day experiment. These findings indicate that glucose-based AnGS exhibits better resistance to TA toxicity. Overall, a TA concentration of 2000 mg·L$^{-1}$ only partially inhibited AnGS with glucose as a carbon source.

As shown in Figure 2, the specific methanogenic activity (SMA) was determined to assess the methanogenic activity of AnGS. In the control reactor without TA addition, the SMA cultured with sodium acetate or glucose was approximately 8.00 mL CH$_4$ (gVSS·h)$^{-1}$, consistent with research findings [38]. At TA concentrations of 500 mg·L$^{-1}$ and 1000 mg·L$^{-1}$,

the SMA of AnGS in STA-I and STA-II remained comparable to the control, suggesting that TA did not inhibit methane-producing activity at these concentrations. However, the SMA of AnGS in GTA-II was slightly smaller than that in the control. The influence of the TA concentration on the SMA of AnGS was influenced by the carbon source substrate. The reduced SMA in GTA-II may be due to TA interacting with glucose, slowing bacterial anaerobic fermentation rates, and reducing the substrate supply for the methanogenic stage. Thus, at TA concentrations below 1000 mg·L$^{-1}$, no significant inhibition in SMA was observed in AnGS. In the sodium acetate group, SMA of STA-III dropped from 7.71 mL CH$_4$ (gVSS·h)$^{-1}$ to zero after TA reached 1500 mg·L$^{-1}$, completely inhibiting methanogenic activity. Likewise, the SMA of STA-IV experienced a sudden decline starting on the 8th day, with methane production nearly ceasing by the 14th day. In the glucose-fed group, the inhibition of SMA by 2000 mg·L$^{-1}$ TA was partial. Furthermore, SMA in the GTA-IV reactor was only 25.13% lower than that in the control throughout the experiment. In conclusion, when TA concentrations reach a threshold, they significantly inhibit methane production in AnGS, leading to VFAs' accumulation and carbon source degradation. It is worth noting that the inhibition of AnGS by TA was more pronounced in the sodium acetate group than in the glucose group with increasing TA concentration.

### 3.2. The Impact of TA on Effluent VFAs Concentration

The content and composition of VFAs in each experimental group were measured (Figure 3). When the concentration of TA was not higher than 1000 mg·L$^{-1}$, the effluent VFA concentrations in the TA-treated reactors remained comparable to those in the corresponding control reactors (Figure 3a,b). At TA concentrations that did not exceed 1000 mg·L$^{-1}$, TA had almost no effect on the conversion of sodium acetate or glucose, with no significant presence of VFA intermediates. However, when TA concentrations were 1500 mg·L$^{-1}$ and 2000 mg·L$^{-1}$, a large accumulation of VFAs in the effluent of STA-III and STA-IV was found. Likewise, the effluent VFA content in GAT-IV was partially accumulated at the TA concentration of 2000 mg·L$^{-1}$. At the same TA level (TA = 2000 mg·L$^{-1}$), the extent of VFA accumulation was more severe in the sodium acetate group than in the glucose group. On the 14th day, the acetate concentration in STA-IV (resulting from sodium acetate and the degradation of TA) exceeded 2000 mg·L$^{-1}$. By the 30th day, it remained stable, with over 98.26% of sodium acetate remaining unutilized by methanogens. To prevent rapid glucose fermentation leading to system acidification, NaHCO$_3$ was introduced as a buffering agent to each reactor within the glucose group in this study. However, after the 20th day, the effluent VFAs in GTA-IV also exceeded 2000 mg·L$^{-1}$ (Figure 3b), and the effluent pH decreased to approximately 6.5 (Figure 1d), indicating acidification in the reactor. Subsequent analysis of the composition of accumulated VFAs revealed that TA inhibited the methanogenic process, primarily resulting in the gradual decline of anaerobic acetic acid degradation by sludge methanogens. On the 35th day, concurrent with the rise in effluent VFA concentration, substantial quantities of propionic acid (~1000 mg·L$^{-1}$) and n-butyric acid (~500 mg·L$^{-1}$) were detected in GTA-IV (Figure 3d). At the same time, the gas production capacity in GTA-IV decreased by 40.36%. This indirectly reflects that the high concentration of TA interferes with the metabolic reactions involved in hydrogen and acetic acid production during the glucose degradation process, specifically inhibiting the acetylation process of propionic acid and butyric acid. The results indicate that when the concentration of TA reaches a threshold, it initially inhibits the methane-producing activity of AnGS and reduces the gas-production capacity of sludge. Subsequently, this leads to the accumulation of acetic acid and further inhibits the hydrogen and acetic acid production stages.

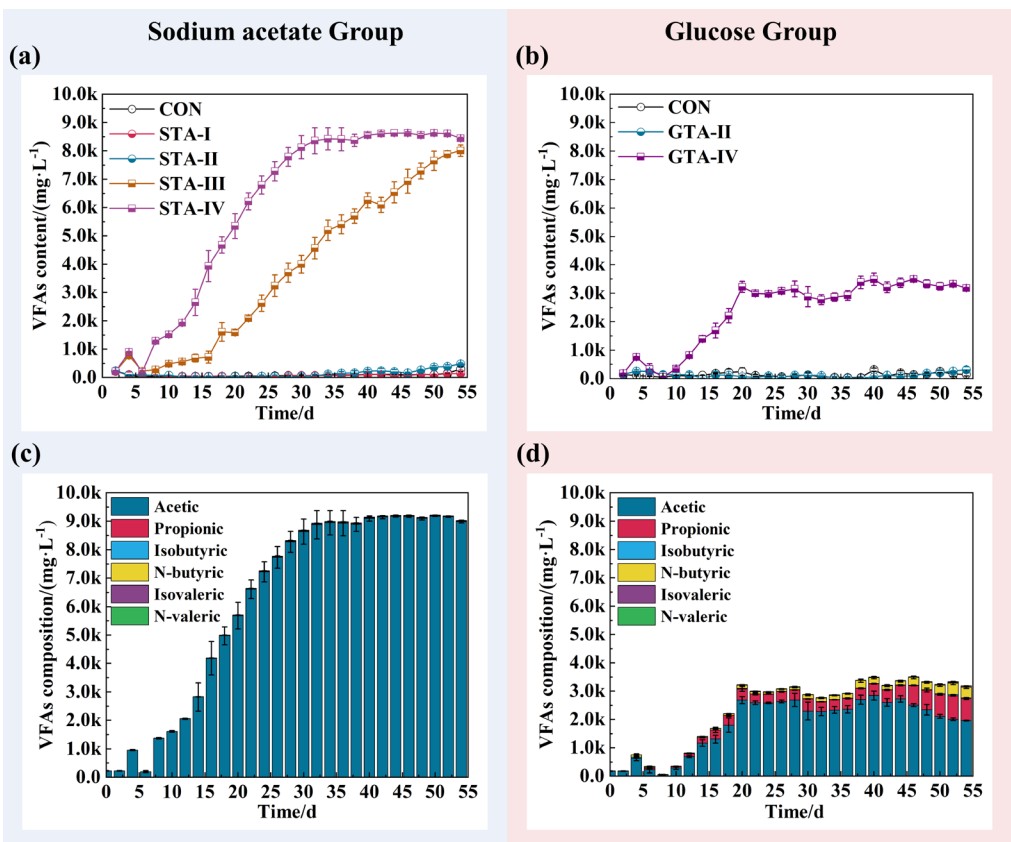

**Figure 3.** The change in content ((**a**) sodium acetate group, (**b**) glucose group) and composition (**c**) sodium acetate group, (**d**) glucose group, TA was 2000 mg·L$^{-1}$) of VFAs during anaerobic digestion under TA stress.

### 3.3. Deflocculation of AnGS by Toxicity of TA

Key evidence demonstrating that TA induces deflocculation and sludge loss was obtained through comparative analyses of variations in AnGS morphology, anaerobic effluent turbidity, and the composition of extracellular polymeric substances (EPS) across different TA concentration levels. Figure 4 illustrates that the addition of TA led to evident rupture and fragmentation of AnGS. Subsequently, the effluent conditions of all reactors were further analyzed. As depicted in Figure S2, in the absence of TA in the influent, the disparity between the effluent samples before and after filtration was negligible. The turbidity of the anaerobic effluent increased gradually with the rise in TA concentration in the influent, and the cloudy effluent became clear and colorless after filtration. This phenomenon suggests that TA weakens the strength of AnGS by interacting with EPS, resulting in the release of a large number of free microbial colloids and increased effluent turbidity. Furthermore, the addition of TA caused significant damage to the structure of the AnGS and notably reduced its overall particle size, as evidenced by particle size distribution analysis (refer to Figure S3). The median particle size (D50) of AnGS was negatively correlated with the concentration of TA, with a more significant association observed in the sodium acetate group ($R > 0.98$, $p < 0.05$).

The protein (PN) and polysaccharide (PS) in TB-EPS and LB-EPS were analyzed following the methodology outlined by Li et al. [30]. For TB-EPS, both the PN and PS contents noticeably decreased as the TA concentration increased in both experimental groups (Figure 5). In the sodium acetate group, the PN and PS contents of TB-EPS decreased by 33.39–49.41% and 20.72–42.11% compared to the control (CON), respectively. Similarly, in the glucose group, the PN and PS contents of TB-EPS decreased by 32.05–41.03% and 22.63–41.82% compared to the control (CON), respectively. Rich functional groups in PN can facilitate connections with other EPS components through complex interactions [39]. It

has been widely reported that TA can bind with PN via hydrogen bonding and hydrophobic interactions [40]. Additionally, PS can form ternary complexes with TA-PN aggregates, enhancing their solubility [41,42]. This alteration affects the state of EPS in the sludge. TA administration resulted in a significant decrease in PN and PS contents in TB-EPS, indicating the effective complexing and precipitating effects of TA on PN. However, the content of PN and PS in LB-EPS noticeably increased as the TA concentration increased in both experimental groups. The PN and PS contents of LB-EPS in the sodium acetate group increased by 444.44–422.22% and 115.67–37.79% compared to the control (CON), respectively. Analogously, the PN and PS contents of LB-EPS in the glucose group increased by 25.20~54.80% and 15.67~37.79% compared to the control (CON), respectively. LB-EPS containing more PN and PS under TA stress may be attributed to TA disrupting the sludge structure by precipitating PN and PS in EPS, thereby exposing the EPS originally encapsulated within the sludge and leading to the conversion of TB-EPS to LB-EPS. The composition and distribution of PN and PS in EPS are crucial for maintaining the stability of AnGS [43]. Therefore, TA might gradually decompose and disrupt granular sludge by weakening the interaction between EPS and AnGS, resulting in biomass loss in anaerobic reactors.

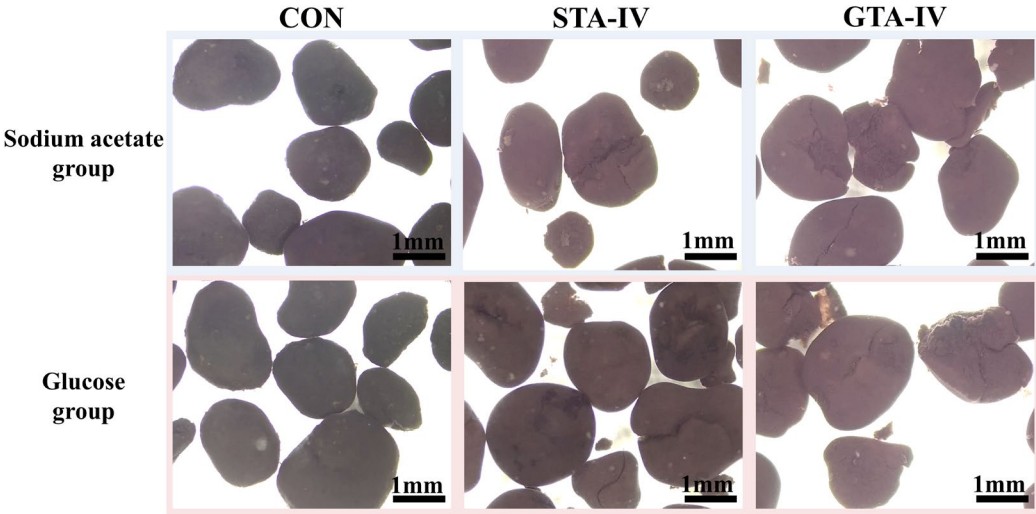

**Figure 4.** Morphological image of AnGS under TA stress.

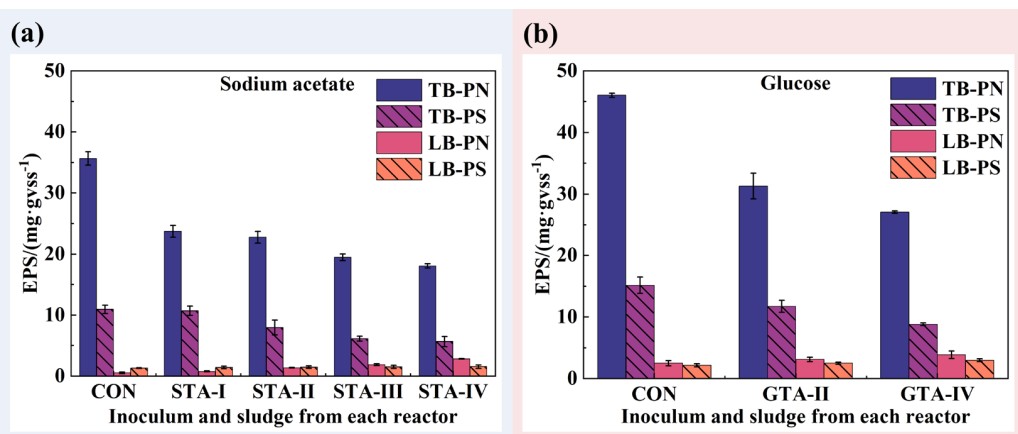

**Figure 5.** Content and composition of PN and PS in EPS of AnGS under TA stress ((**a**) sodium acetate group, (**b**) glucose group).

*3.4. Microbial Response to TA Stress*

3.4.1. Microbial Diversity

Microorganisms play a pivotal role in anaerobic digestion under TA stress. As demonstrated in Table 3, the addition of TA resulted in a significant reduction in bacterial richness and diversity. In the sodium acetate group, the Shannon index, which positively correlates with microbial diversity [44], increased from 3.797 in the control (CON) to 1.901 in the STA-IV reactor. Meanwhile, the Chao 1 index, an indicator of microbial richness [45], increased from 708.890 in CON to 555.362 in STA-IV. Similar trends were observed in the glucose group. Additionally, the Venn diagram (Figure 6a,b) illustrates the effect of TA concentrations on changes in bacterial composition. In the sodium acetate group, the number of OTUs unique to bacteria decreased from 63 in CON to 14 in STA-IV. Similarly, in the glucose group, the number of unique OTUs for bacteria decreased from 187 to 19. In conclusion, bacterial communities exhibited consistent changes with increasing TA concentrations.

**Table 3.** Bacteria community diversity analysis.

| Index Type | Inoculum | Sodium Acetate | | | | | Glucose | | |
| --- | --- | --- | --- | --- | --- | --- | --- | --- | --- |
| | | **CON** | **STA-I** | **STA-II** | **STA-III** | **STA-IV** | **CON** | **GTA-II** | **GTA-IV** |
| Coverage | 0.995 | 0.998 | 0.998 | 0.998 | 0.998 | 0.998 | 0.998 | 0.998 | 0.999 |
| Chao 1 | 1178.665 | 708.890 | 693.746 | 596.611 | 588.622 | 555.362 | 678.079 | 573.907 | 320.279 |
| Pielou | 0.577 | 0.588 | 0.545 | 0.437 | 0.440 | 0.314 | 0.586 | 0.518 | 0.355 |
| Shannon | 4.005 | 3.797 | 3.515 | 2.743 | 2.644 | 1.901 | 3.755 | 3.214 | 1.967 |

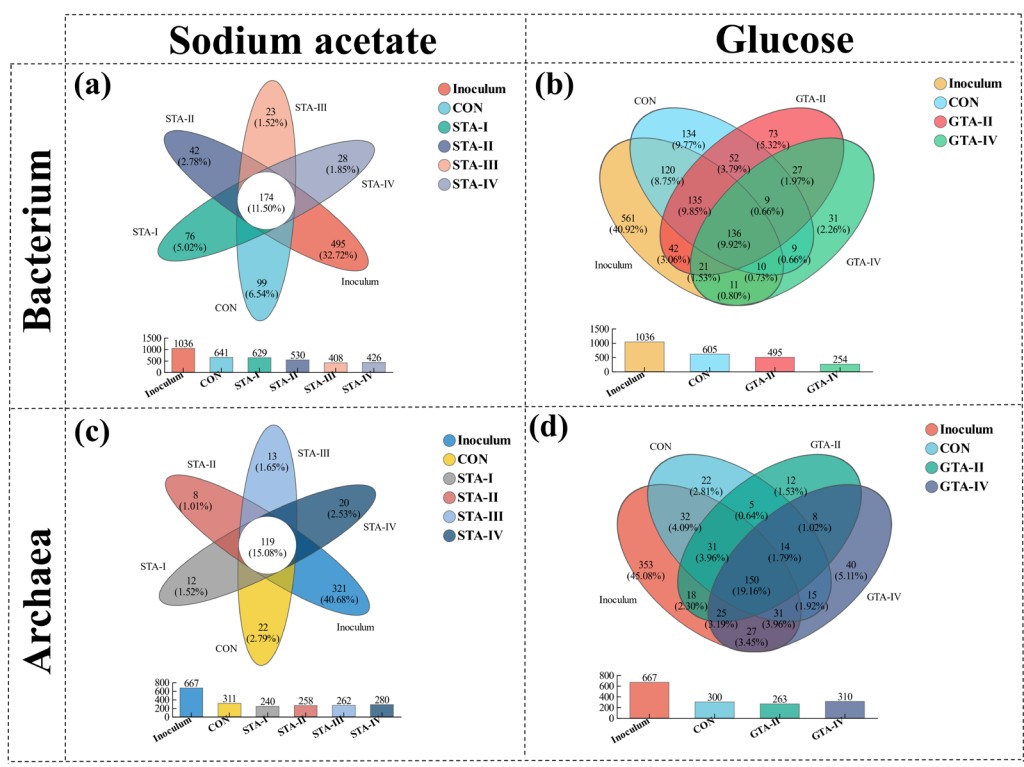

**Figure 6.** Venn diagrams of bacteria ((**a**) sodium acetate substrate, (**b**) glucose substrate) and archaea ((**c**) sodium acetate substrate, (**d**) glucose substrate) in AnGS at OUTs level.

Conversely, archaeal diversity showed a positive correlation with TA concentrations in both the sodium acetate and glucose groups. If the TA concentration was below the threshold (1500–2000 mg·L$^{-1}$), as indicated in Table 4, the archaeal Shannon values were lower in GTA-I and GTA-II compared to the control (CON). However, when TA concentrations exceeded this threshold, the archaeal Shannon values were higher in GTA-III and GTA-IV

than in the control (CON). The differences in the number of OTUs unique to archaea were also analyzed. As the TA concentrations increased, the number of OTUs unique to archaea rose from 12 to 20 in the sodium acetate group and from 12 to 40 in the glucose group (Figure 6c,d). TA, serving as a metabolizable co-substrate, can alter the nutrient profile of archaea following hydrolysis and acidification. Consequently, elevated TA concentrations promote species diversity in archaea by enriching and enhancing substrate composition.

**Table 4.** Archaeal community diversity analysis.

| Index Type | Inoculum | Sodium Acetate | | | | | Glucose | | |
|---|---|---|---|---|---|---|---|---|---|
| | | CON | STA-I | STA-II | STA-III | STA-IV | CON | GTA-II | GTA-IV |
| Coverage | 0.993 | 0.998 | 0.998 | 0.998 | 0.998 | 0.998 | 0.998 | 0.998 | 0.998 |
| Chao 1 | 937.234 | 433.526 | 312.857 | 328.222 | 357.067 | 346.978 | 372.226 | 320.130 | 376.500 |
| Pielou | 0.299 | 0.284 | 0.192 | 0.223 | 0.276 | 0.301 | 0.325 | 0.330 | 0.347 |
| Shannon | 1.946 | 1.630 | 1.052 | 1.237 | 1.535 | 1.694 | 1.856 | 1.839 | 1.993 |

3.4.2. Bacterial Community

The analysis of bacterial communities, as depicted in Figure 7, was conducted. At the phylum level, *Firmicutes* and *Chloroflexi* were predominant in both groups (Figure 7a,c). Additionally, *Actinobacteriota* was the second most dominant phylum after *Firmicutes* in the glucose group. Similarly, the genera *unclassified_f__Lactospiraceae* and *Streptococcus* were the most predominant bacteria in the sodium acetate group (Figure 7b). However, the predominant bacterial genera in the glucose group exhibited irregularities across the sludge obtained from each reactor (Figure 7d). Previous studies have reported that the type of carbon source substrate can influence bacterial composition [46,47]. Nevertheless, this study primarily focused on the effect of TA concentration on bacterial communities.

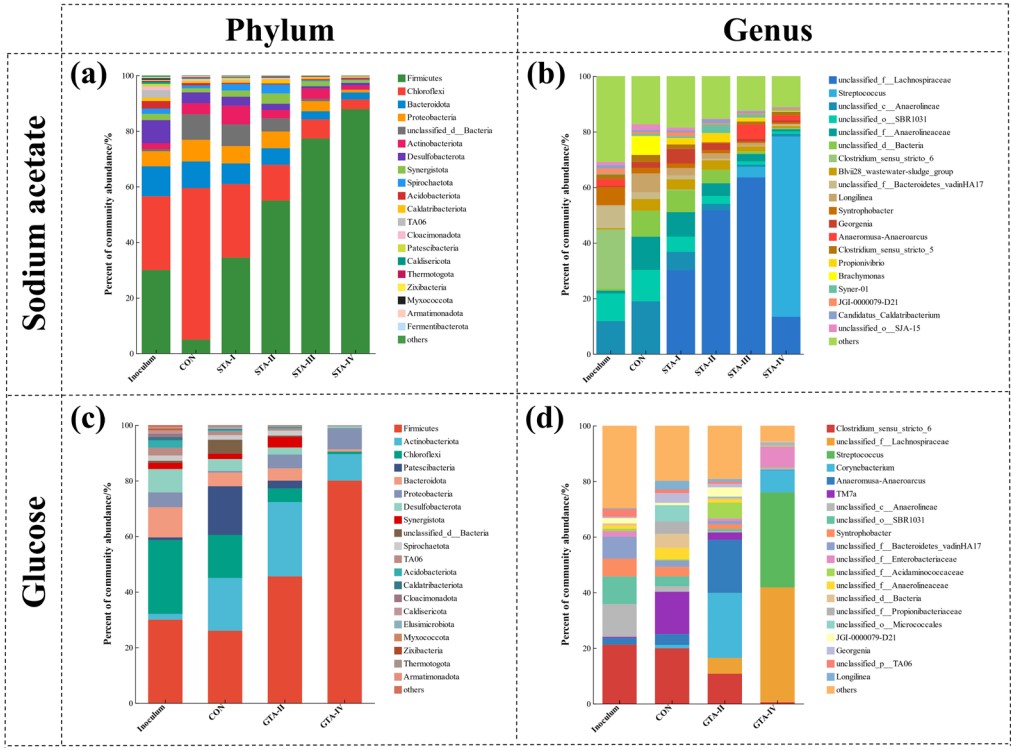

**Figure 7.** Visual histograms of bacteria at the phylum ((**a**) sodium acetate substrate, (**c**) glucose substrate) and genus ((**b**) sodium acetate substrate, (**d**) glucose substrate) levels show the distribution of different species in each reactor under TA stress proportions and relationships.

The relative abundance of the phylum *Firmicutes*, which comprises common hydrolytic-acid-producing bacteria in anaerobic reactors, was found to be positively correlated with TA concentration, particularly evident in the sodium acetate group ($R > 0.98$, $p < 0.05$). At the genus level, *unclassified_f__Lactospiraceae* and *Streptococcus*, both belonging to the phylum *Firmicutes*, are primarily involved in carbohydrate fermentation, producing acetic acid, propionic acid, and butyric acid for methane synthesis [48–50]. When the TA concentration did not exceed 1500 mg·L$^{-1}$, the genus *unclassified_f__Lachnospiraceae* dominated in the sodium acetate group. Furthermore, the relative abundance of *unclassified_f__Lachnospiraceae* increased with higher TA concentrations. However, when the TA concentrations reached 2000 mg·L$^{-1}$, the relative abundance of *unclassified_f__Lachnospiracea* dropped rapidly in STA-IV. *Streptococcus* replaced *unclassified_f__Lactospiraceae* as the dominant bacterial genus. When the TA concentration reached 2000 mg·L$^{-1}$, *unclassified_f__Lactospiraceae* and *Streptococcus* also emerged as the predominant bacterial genera in GTA-IV. This observation confirms the convergence of bacterial composition in the sludge due to the aforementioned increase in TA concentration.

However, the relative abundance of the phylum *Chloroflexi* was found to be negatively correlated with TA concentration, particularly evident in the sodium acetate group ($R < -0.92$, $p < 0.05$). *Chloroflexi* is commonly observed in anaerobic reactors [51]. Characterized by multiple nutritional modes, *Chloroflexi* can enhance biofilm formation by secreting proteins, thereby increasing the stability of granular sludge [52,53]. The observed reduction in protein and polysaccharide contents in sludge EPS (Figure 5), fragmentation of granular sludge (Figure 4), and loss of flocculated sludge (Figure S2) in this experiment may be attributed to the inhibition of *Chloroflexi* function by TA.

Of note, noticeable variations in TA concentration exerted distinct impacts on the relative abundance of the phylum *Actinobacteriota* in both experimental groups. Prior to and following the attainment of the TA threshold concentration, we observed an initial increase followed by a subsequent decrease in the relative abundance of *Actinobacteriota* in both the sodium acetate and glucose groups. This threshold concentration is influenced by the type of carbon source, with values ranging from 500–1000 mg·L$^{-1}$ in the sodium acetate group to 1000–2000 mg·L$^{-1}$ in the glucose group. *Actinobacteriota* plays a pivotal role in the decomposition of complex polymers, thereby maintaining the balance of anaerobic digestion systems [54]. When the TA concentration exceeds the specific threshold, it may destabilize the anaerobic sludge system by inhibiting *Actinobacteriota* enrichment.

Moreover, with increasing TA concentration, the relative abundance of the genus *Clostridium_sensu_stricto_6* decreased. *Clostridium_sensu_stricto_6* is involved in organic matter degradation, leading to the production of VFAs [55]. Similarly, the relative abundance of *unclassified_f_Bacteroidetes_vadinHA17* also decreased with increasing TA concentration. *Unclassified_f_Bacteroidetes_vadinHA17* is capable of fermenting glucose to produce propionate, acetate, and $H_2/CO_2$ [56,57]. The accumulation of propionic acid and butyric acid observed in this experiment (Figure 3d) may be attributed to the decreased activity of these specific bacterial genera under high TA concentrations.

### 3.4.3. Archaeal Community

The presence of the *mcrA* gene serves as an indicator of methanogen content, facilitating the evaluation of methanogenic activity in sludge [15]. Figure 8 illustrates the correlation between TA concentrations and *mcrA* gene content. In the sodium acetate group, *mcrA* gene content ranged from $7.39 \times 10^{10}$ copies·gVSS$^{-1}$ in the control (CON) to $1.07 \times 10^{10}$ copies·gVSS$^{-1}$ in STA-IV. Importantly, these variations occurred when TA concentrations remained below 1000 mg·L$^{-1}$. The relationship between *mcrA* gene content and specific methanogenic activity (SMA) within the sodium acetate group remained relatively consistent. Consequently, there was a slight increase in sludge methanogenic activity, while gas generation remained stable. However, when TA concentration exceeded 1500 mg·L$^{-1}$, *mcrA* gene content decreased dramatically to $1.16 \times 10^{10}$ copies·gVSS$^{-1}$ in STA-III and $1.25 \times 10^{10}$ copies·gVSS$^{-1}$ in STA-IV, indicating a significant inhibition of

methanogenic activity, to the extent that activity was completely lost (Figure 2). In the glucose group, *mcrA* gene content decreased from $6.49 \times 10^{10}$ copies·gVSS$^{-1}$ in the control (CON) to $5.20 \times 10^{10}$ copies·gVSS$^{-1}$ in GTA-IV as the TA concentration increased. This decline corresponded with changes observed in methanogenic activity and gas production. Overall, a clear and persistent correlation was observed between *mcrA* gene content and the degree of specific methanogenic activity under TA stress.

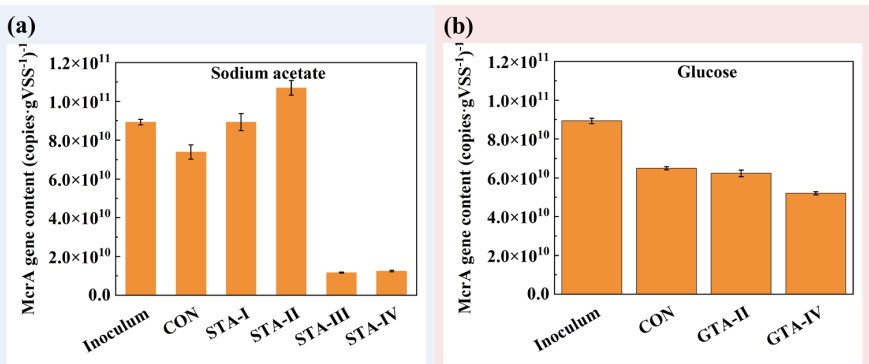

**Figure 8.** Level of methanogenesis-related *mcrA* gene in AnGS from each reactor under TA stress ((**a**) sodium acetate group, (**b**) glucose group).

To explore the effects of TA stress on the methanogenic community, we conducted a detailed analysis of the archaeal composition in the sludge (Figure 9). In both the sodium acetate and glucose groups, *Halobacterota* and *Euryarchaeota* were identified as the predominant archaeal phyla in the anaerobic digestion systems (Figure 9a,c). The prevailing archaeal genera were *Methanosaeta* (an aceticlastic methanogen) and *Methanobacterium* (a hydrogenotrophic methanogen) (Figure 9b,d) [58].

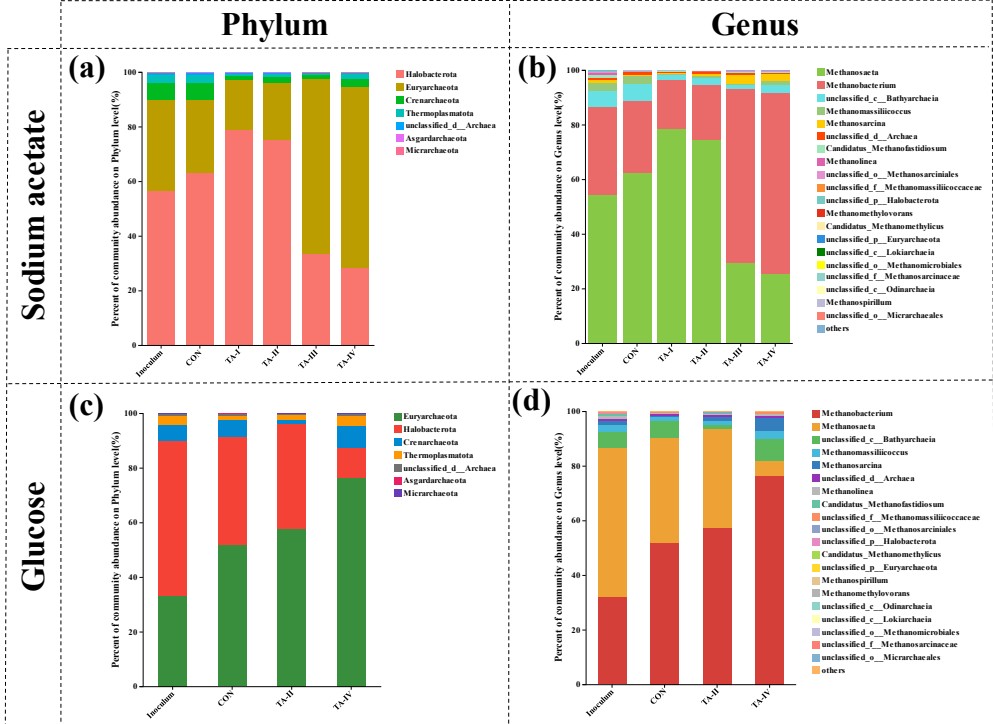

**Figure 9.** Visual histograms of archaea at the phylum ((**a**) sodium acetate substrate, (**c**) glucose substrate) and genus ((**b**) sodium acetate substrate, (**d**) glucose substrate) levels show the distribution of different species in each reactor under TA stress proportions and relationships.

The archaeal composition in the two groups exhibited less variability when the TA concentrations were below 1000 mg·L$^{-1}$. In both groups, the relative abundance of *Methanosaeta* was slightly higher in the experimental reactors (STA-I, STA-II, and GTA-II) compared to the control reactors (CON). This difference can be attributed to the conversion of TA to acetic acid during anaerobic digestion, thereby promoting the proliferation of *Methanosaeta*. However, under high TA concentrations (>1500 mg·L$^{-1}$), significant changes occurred in the archaeal composition. The relative abundance of *Methanosaeta* sharply declined, while that of *Methanobacterium* increased. Consequently, TA reduced methane production (Figure 1c,f) and resulted in the accumulation of acetic acid (Figure 3c,d), possibly due to decreased *Methanosaeta* activity. Notably, the abundance of *Methanosarcina*, another aceticlastic methanogenic archaeal genus, exhibited a positive correlation with TA concentrations. *Methanosarcina* demonstrates high environmental adaptability and resistance, as it can utilize H$_2$ and CO$_2$ to generate methane, in addition to the acetate methanogenic pathway [59]. Furthermore, the abundance of other archaeal genera, all of which are hydrogenotrophic methanogens, increased with increasing TA concentrations. Therefore, hydrogenotrophic methanogenic archaea demonstrate greater adaptability than aceticlastic methanogenic archaea under high TA concentrations (>1500 mg·L$^{-1}$).

## 4. Conclusions

The methanogenic activity, morphology, and microbial community in granule-based anaerobic reactors fed with sodium acetate and glucose in the presence of TA were studied. The results showed that the methanogenic activity was slightly affected by TA stress when the TA concentration was lower than 1000 mg·L$^{-1}$. However, TA concentrations above 1500 mg·L$^{-1}$ led to a significant depression of methanogenic activity for the sodium acetate group, and bulk pH dropped below 7.0. In the sodium acetate group, TA-induced inhibition occurred when the TA concentration exceeded 1500 mg·L$^{-1}$, resulting in the excessive accumulation of acetic acid. At a TA concentration of 2000 mg·L$^{-1}$, TA-induced inhibition was only partially observed in the glucose group. The presence of TA changed the EPS content and composition, which might weaken the stability of granular sludge and cause biomass loss from the sludge. On the other hand, bacterial diversity was reduced by TA stress, and archaea showed an opposite change. The antitonicity of hydrogenotrophic methanogens among the archaea was stronger than that of aceticlastic methanogens in the TA-affected anaerobic reactors. This study provides informative results to access and analyze the responses of the AnGS-based reactor to plant polyphenol-containing wastewater. Using TA as a sole model compound to investigate the impact of tannin-containing wastewater on anaerobic granular sludge may present a biased perspective. Therefore, further exploration is needed to determine whether the inhibitory forms of tannin-containing wastewater in actual anaerobic treatment are consistent with the inhibition mechanisms induced by TA.

**Supplementary Materials:** The following supporting information can be downloaded at: https://www.mdpi.com/article/10.3390/fermentation10050262/s1, Text S1: The activation of inoculum; Text S2: Detailed description of the conventional analysis methods; Text S3: Detailed description of statistical analysis; Figure S1: The content of TA in anaerobic effluent under TA stress (a: sodium acetate group; b: glucose group); Figure S2: Turbidity image of anaerobically digested effluent under TA stress; Figure S3: Particle size distribution of AnGS at the end of the experiment under TA stress (a: sodium acetate group; b: glucose group).

**Author Contributions:** Conceptualization, S.B.; data curation, S.B. and H.L.; formal analysis, S.B., H.L. and H.Z.; investigation, S.B.; methodology, S.B.; resources, J.Z.; software, Z.L.; supervision, Y.C. and J.Z.; validation, H.Z.; visualization, S.B.; writing—original draft preparation, S.B. and H.L.; writing—review and editing, S.B., H.L. and J.Z. All authors have read and agreed to the published version of the manuscript.

**Funding:** This research was supported by the National Key Research and Development Program (2022YFC2105505), the National Natural Science Foundation of China (22208066), the Guangxi Natural

Science Foundation of China (2023GXNSFGA026001), and the Nanning Innovation and Entrepreneur Leading Talent Project (2021001).

**Institutional Review Board Statement:** Not applicable.

**Informed Consent Statement:** Not applicable.

**Data Availability Statement:** The data presented in this study are available within the article and the Supplementary Materials.

**Conflicts of Interest:** Author Jian Zhang was employed by the company Anhui Bossco Environmental Protection Technology Co., Ltd. The remaining authors declare that the research was conducted in the absence of any commercial or financial relationships that could be construed as a potential conflict of interest.

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
