# Peer review of "Response of Anaerobic Granular Sludge Reactor to Plant Polyphenol Stress: Floc Disintegration and Microbial Inhibition"

_fermentation, doi:10.3390/fermentation10050262_

Round 1

Reviewer 1 Report

Comments and Suggestions for Authors

Dear authors, 

The manuscript titled "Response of anaerobic granular sludge reactor to plant polyphenols stress:Floc disintegration and microbial inhibition" appears to have valuable content, but there are areas where improvement can be made to enhance its suitability for publication. Here are some suggestions for improvement:

Abstract

Rewrite the following sentence ( The content of the mcr_A gene in granular sludge decreased markedly in response to TA stress, providing direct evidence that a high concentration of TA was inhibited).

Line 22, it s not aim, it should be conclusion or recommendation.

Arrange keywords in alphabetic order.

Improve the abstract by adding more details and focus on providing a clear and brief overview of the objectives, methods, results, and conclusions of the study.

Introduction

Line 48, What is F420?

Line 51-53, long sentence it should be rewritten.

Add more information about mcr_A genes?

Improve the aim of study and explore more the your research gap. 

Materials and methods,

The authors may add a section of chemicals used and their source. 

Lines 78, what is the artifical wastwater used?

Line 80, what is COD?

In table 1, add the fullname of each term. 

Discussion

Stay focused on the main findings of the study and avoid introducing too much background information.

Discuss the results in the context of existing literature and explain the mechanisms behind the observed effects.

Acknowledge the limitations of the study and suggest future research directions.

In conclusion Briefly summarize the main findings of the study in a clear and concise way.

Comments to language 

Some sentences can be shortened or rephrased to be more concise. 

 Proofread carefully for typos and grammatical errors.

References 

Please search for more recent references. 

Comments on the Quality of English Language

Some sentences can be shortened or rephrased to be more concise. 

 Proofread carefully for typos and grammatical errors.

Author Response

请参阅附件。

Reviewer 2 Report

Comments and Suggestions for Authors

Fermentation-2982089

Response of anaerobic granular sludge reactor to plant polyphenols stress:Floc disintegration and microbial inhibition

Review Comments:

First impressions

-       The authors need to work with a native English speaking editor to re-write this manuscript.

-       The scientific merits of this paper cannot be assessed until the Technical English has been corrected..  

Abstract

-       The abstract presents a good summary of the ideas and outcomes described in the manuscript.

Corrections: 

Page 1, lines 20-23 The content of the mcr_A gene in granular sludge decreased markedly in response to TA stress, providing direct evidence that a high concentration of TA was inhibited.” Change to The concentration of the mcr_A gene in granular sludge decreased markedly in response to TA stress, providing direct evidence that a high concentration of TA caused inhibition of specific gene expressions.

Introduction

Corrections:

Page 1, lines 37-39 The methanogenic toxicity of TA was reported to depend on concentration, and its degradation would constantly cause metabolic depression until completely disappeared.” Change to “The methanogenic toxicity of TA was reported to depend on concentration, and its degradation would constantly cause metabolic depression until the TA was completely degraded.”

Page 1, line 39-41 Because of negative impacts on enzyme activity by creating hydrogen bonds with the active areas of the enzyme [8,9], so the high initial concentration could directly inhibit the activity of sludge.”  This is not a sentence.

Page 1-2, lines 44-46 The ability to bind to enzymes of tannin monomers was weaker than oligomeric tannins, while the phenolic hydroxyl groups buried within the molecule of poly-tannins are difficult to contact with enzymes”  re-write…..multiple mistakes regarding subject, predicate object, etc…….

Page 2, lines 47-48   it was reported that F420 content in methanogenic bacterial cells decreases with increasing tannins”  Identify and/or describe F420

:::::::::::::::::::::::::::::::::::::::::::::::::::::::::::::::::::::::::::::::::::::::::::::::::::::

TOO MANY MISTAKES IN BASIC TECHNICAL ENGLISH

REVIEW HALTED UNTIL NEXT RE-WRITE

::::::::::::::::::::::::::::::::::::::::::::::::::::::::::::::::::::::::::::::::::::::::::::::::::::

Materials and Methods

Corrections:

Results and Discussion

Figures and Tables

Corrections:

Comments/Suggestions for future experiments or analyses:

Conclusion

Corrections:

References, tables and figures

Should this manuscript be published?

NOTE REVIEWERS FOR THIS JOURNAL PROVIDE THEIR SERVICES FOR FREE. DO NOT WASTE OUR TIME WITH POORLY WRITTEN MANUSCRIPTS. 

Comments on the Quality of English Language

Fermentation-2982089

Response of anaerobic granular sludge reactor to plant polyphenols stress:Floc disintegration and microbial inhibition

Review Comments:

First impressions

-       The authors need to work with a native English speaking editor to re-write this manuscript.

-       The scientific merits of this paper cannot be assessed until the Technical English has been corrected..  

Abstract

-       The abstract presents a good summary of the ideas and outcomes described in the manuscript.

Corrections: 

Page 1, lines 20-23 The content of the mcr_A gene in granular sludge decreased markedly in response to TA stress, providing direct evidence that a high concentration of TA was inhibited.” Change to The concentration of the mcr_A gene in granular sludge decreased markedly in response to TA stress, providing direct evidence that a high concentration of TA caused inhibition of specific gene expressions.

Introduction

Corrections:

Page 1, lines 37-39 The methanogenic toxicity of TA was reported to depend on concentration, and its degradation would constantly cause metabolic depression until completely disappeared.” Change to “The methanogenic toxicity of TA was reported to depend on concentration, and its degradation would constantly cause metabolic depression until the TA was completely degraded.”

Page 1, line 39-41 Because of negative impacts on enzyme activity by creating hydrogen bonds with the active areas of the enzyme [8,9], so the high initial concentration could directly inhibit the activity of sludge.”  This is not a sentence.

Page 1-2, lines 44-46 The ability to bind to enzymes of tannin monomers was weaker than oligomeric tannins, while the phenolic hydroxyl groups buried within the molecule of poly-tannins are difficult to contact with enzymes”  re-write…..multiple mistakes regarding subject, predicate object, etc…….

Page 2, lines 47-48   it was reported that F420 content in methanogenic bacterial cells decreases with increasing tannins”  Identify and/or describe F420

:::::::::::::::::::::::::::::::::::::::::::::::::::::::::::::::::::::::::::::::::::::::::::::::::::::

TOO MANY MISTAKES IN BASIC TECHNICAL ENGLISH

REVIEW HALTED UNTIL NEXT RE-WRITE

::::::::::::::::::::::::::::::::::::::::::::::::::::::::::::::::::::::::::::::::::::::::::::::::::::

Materials and Methods

Corrections:

Results and Discussion

Figures and Tables

Corrections:

Comments/Suggestions for future experiments or analyses:

Conclusion

Corrections:

References, tables and figures

Should this manuscript be published?

NOTE REVIEWERS FOR THIS JOURNAL PROVIDE THEIR SERVICES FOR FREE. DO NOT WASTE OUR TIME WITH POORLY WRITTEN MANUSCRIPTS. 

Author Response

请参阅附件。

Round 2

Reviewer 1 Report

Comments and Suggestions for Authors

No further comments

Reviewer 2 Report

Comments and Suggestions for Authors

Fermentation-2982089 v2

Response of anaerobic granular sludge reactor to plant polyphenols stress: Floc disintegration and microbial inhibition

Review Comments:

First impressions

-       The authors need to work with a native English speaking editor to re-write this manuscript. CORRECTED

-       The scientific merits of this paper cannot be assessed until the Technical English has been corrected.. CORRECTED

-       The authors have made a substantial improvement in the quality of the manuscript.

Abstract

-       The abstract presents a good summary of the ideas and outcomes described in the manuscript.

Corrections:  No corrections are needed in this part of the manuscript.

Page 1, lines 20-23 The content of the mcr_A gene in granular sludge decreased markedly in response to TA stress, providing direct evidence that a high concentration of TA was inhibited.” Change to The concentration of the mcr_A gene in granular sludge decreased markedly in response to TA stress, providing direct evidence that a high concentration of TA caused inhibition of specific gene expressions. CORRECTED

Introduction

Corrections:

Page 1, lines 37-39 The methanogenic toxicity of TA was reported to depend on concentration, and its degradation would constantly cause metabolic depression until completely disappeared.” Change to “The methanogenic toxicity of TA was reported to depend on concentration, and its degradation would constantly cause metabolic depression until the TA was completely degraded.” CORRECTED

Page 1, line 39-41 Because of negative impacts on enzyme activity by creating hydrogen bonds with the active areas of the enzyme [8,9], so the high initial concentration could directly inhibit the activity of sludge.”  This is not a sentence.

CORRECTED

Page 1-2, lines 44-46 The ability to bind to enzymes of tannin monomers was weaker than oligomeric 44 tannins, while the phenolic hydroxyl groups buried within the molecule of poly-tannins are difficult to contact with enzymes”  re-write…..multiple mistakes regarding subject, predicate object, etc……. CORRECTED

Page 2, lines 47-48   it was reported that F420 content in methanogenic bacterial cells decreases with increasing tannins”  Identify and/or describe F420 CORRECTED

-       The authors have provided a good overview of the role that polyphenols can have on anaerobic digestion.

-       The authors have pointed out this manuscripts unique contribution to this field of study.

Materials and Methods

-       The methodology section describes each of the experimental design steps, water quality analysis and molecular methods in depth such that the analysis could be reproduced by another researcher.

-       The authors have done a good job of describing their methodology.

Corrections: No corrections are needed in this part of the manuscript.

Results and Discussion

-       The results and discussion sections are well written.

-       The authors present a good case for the interpretations and conclusions based on the genetic data and geographic locations.

-       The authors are to be commended for connecting all the aspects of microbial processes (water quality, sludge granule size, polysaccharide and protein content, microbial community analysis and qPCR-gene analysis) and how they are influenced by the presence of plant polyphenols

Figures and Tables

- The figures and tables are easy to read and impart a good deal of information.

Corrections: No corrections are needed in this part of the manuscript.

Comments/Suggestions for future experiments or analyses:

-       There are no additional experiments or analyses needed for this manuscript.

-       In a future series of experiments, it may be interesting to look at pre-treating the industrial wastewater with ozone to reduce the concentration and structural complexity of polyphenols such that they do not inhibit methanogenesis.

Conclusion

-       The conclusions are in line with the data generated by the authors.

-       There are no unjustified claims nor any need to tone down the conclusions.

-       There are no redundancies between the text and the figures and tables.

Corrections: No corrections are needed in this part of the manuscript.

References, tables and figures

-       The citations are appropriate and accurate.

-       The figures and tables need minor corrections.

-       Due to the complexity of the figures, there is a definite need for color in the figures.

Should this manuscript be published?

-       This manuscript is ready for publication.